# Bounds on the entanglement entropy by the number entropy in non-interacting fermionic systems

**Maximilian Kiefer-Emmanouilidis[1,2], Razmik Unanyan[1], Jesko Sirker [2⋆] and Michael Fleischhauer [1]**

**1** Department of Physics and Research Center OPTIMAS,
University of Kaiserslautern, 67663 Kaiserslautern, Germany
**2** Department of Physics and Astronomy, University of Manitoba, Winnipeg R3T 2N2, Canada

⋆ sirker@physics.umanitoba.ca

## Abstract

Entanglement in a pure state of a many-body system can be characterized by the Rényi entropies $S^{(\alpha)} = \ln \text{tr}(\rho^\alpha)/(1-\alpha)$ of the reduced density matrix $\rho$ of a subsystem. These entropies are, however, difficult to access experimentally and can typically be determined for small systems only. Here we show that for free fermionic systems in a Gaussian state and with particle number conservation, $S^{(2)}$ can be tightly bound—from above and below—by the much easier accessible Rényi number entropy $S_N^{(2)} = -\ln \sum_n p^2(n)$ which is a function of the probability distribution $p(n)$ of the total particle number in the considered subsystem only. A dynamical growth in entanglement, in particular, is therefore always accompanied by a growth—albeit logarithmically slower—of the number entropy. We illustrate this relation by presenting numerical results for quenches in non-interacting one-dimensional lattice models including disorder-free, Anderson-localized, and critical systems with off-diagonal (bond) disorder.



# 1 Introduction

The entanglement between two parts of a many-body system in a pure state can be characterized by the Rényi entropies $S^{(\alpha)} = \ln \text{tr}(\rho^\alpha)/(1-\alpha)$. The von-Neumann entanglement entropy is given by $S = S^{(1)} \equiv \lim_{\alpha \to 1} S^{(\alpha)} = -\text{tr}(\rho \ln \rho)$. Their time evolution, for example following a quantum quench, offers important insights into the dynamics of the many-body system. In very small systems the reduced density matrix $\rho$ can be obtained from quantum-state tomography [1]. In larger systems the Rényi entropies are difficult to access experimentally and rather involved techniques using e.g. multiple copies of a system have to be invoked [2–6]. For systems with particle number conservation, the Rényi entropies can be expressed as

$$S^{(\alpha)} = \frac{1}{1-\alpha} \ln \left( \sum_n p^\alpha(n) \, \text{tr} \tilde{\rho}^\alpha(n) \right) = S_N^{(\alpha)} + S_{\text{conf}}^{(\alpha)}. \tag{1}$$

Here $\rho(n)$ is the block of the reduced density matrix with fixed particle number $n$ and $p(n) = \text{tr}\rho(n)$ is the probability distribution of the particle number $n$ in the partition. $\tilde{\rho}(n) = \rho(n)/\text{tr}\rho(n)$ is the corresponding normalized reduced density matrix. The Rényi number entropy $S_N^{(\alpha)} = \ln[\sum_n p^\alpha(n)]/(1-\alpha)$ — also sometimes called the Rényi generalization of the Shannon entropy or classical Rényi entropy [7,8] — is then the part of the entanglement due to number fluctuations only (i.e., in a system where only one configuration for each possible particle number $n$ exists we would have $S^{(\alpha)} = S_N^{(\alpha)}$) and $S_{\text{conf}}^{(\alpha)}$ describes the additional entanglement due to the existence of several configurations for a given $n$. This configurational entropy takes the particularly simple form, $S_{\text{conf}} = -\sum_n p(n) \text{tr}[\rho(n) \ln \rho(n)]$, in the limit $\alpha \to 1$ [9]. The corresponding number entropy $S_N$ has been measured very recently in an experiment on a cold atomic gas [10]. The source of the number entropy are fluctuations induced by particle transport. $S_{\text{conf}}$, on the other hand, is determined by the full microscopic counting statistics. $S_N$ is thus much easier measurable in experiments and can also be accessed theoretically using conformal field theory (CFT) [11–14].

Here we prove that for non-interacting fermions on a lattice where the particle number is conserved, the second Rényi entropy can be bounded from above *and* below by the second Rényi number entropy $S_N^{(2)}$:

$$\frac{1}{e\pi} \exp\left(2S_N^{(2)}\right) - \frac{1}{6} \lesssim S^{(2)} \lesssim \frac{\ln 2}{\pi} \exp\left(2S_N^{(2)}\right). \tag{2}$$

The existence of strict upper and lower bounds for the entanglement entropy by the same quantity $S_N^{(2)}$ has an important consequence: The size and time dependence of the entanglement is directly linked to that of the number entropy. On the one hand, this allows to measure entanglement through the experimentally much easier accessible number entropy. On the other hand, a dynamical growth of entanglement in any non-interacting fermion system implies that the number entropy grows as well, albeit logarithmically slower. Vice versa, in a fully localized phase, where the fluctuations of the particle number of a partition are expected to saturate, the number of accessible configurations and thus entanglement can no longer increase either.

The relation between entanglement and number fluctuations in a partition of a many-body system has a long history [15–20]. E.g. an exact asymptotic expression of the entanglement entropy in terms of number cumulants has been given in [15], and often the lowest order term provides already a good approximation. Being an asymptotic expansion, convergence is not guaranteed however. Furthermore any truncation of the expansion is neither a strict lower nor a strict upper bound. While strict bounds on the entanglement entropy in terms of the number fluctuations can be given for non-interacting fermions, the bounds in terms of the number entropy in Eq. (2) not only have a clear physical interpretation, we also found

numerical evidence that they carry over to the case of interacting particles [21]. This has important implications for phenomena such as many-body localization.

Our paper is organized as follows. In Sec. 2 we present the proof for the lower and upper bounds in Eq. (2). In Sec. 3 we exemplify the usefulness of these bounds based on numerical data for the time evolution of the entropies after quantum quenches in one-dimensional fermionic lattice models with and without disorder. This includes, in particular, the interesting case of off-diagonal disorder (bond disorder), where the von Neumann entropy shows a very slow $\ln \ln t$ increase in time [22–25], while the number entropy scales as $\ln \ln \ln t$. Extensions to fermionic models with interactions are discussed elsewhere [21].

## 2 Bounds on the Rényi entropy by the number entropy

In the following, we will establish a relation between the second Rényi entropy $S^{(2)} = -\ln \text{tr}(\rho^2)$ of a quantum state $\rho$, which we will refer to as purity entropy, and the corresponding number entropy $S_N^{(2)} = -\ln \sum_n p^2(n)$. Specifically, we consider models of non-interacting fermions with particle number conservation.

### 2.1 Lower bound on the purity entropy

Since the number entropy does not account for the different configurations of particles in the considered subsystem, a trivial lower bound for the purity entropy is given by

$$S^{(2)} \geq S_N^{(2)}. \tag{3}$$

This is, however, in most cases only a very weak bound. An alternative and often much better lower bound can be obtained using the relation between $S^{(2)}$ and the particle number fluctuations $\Delta n^2$ derived in [26]

$$4\ln(2)\Delta n^2 \geq S^{(2)} \geq 2\Delta n^2. \tag{4}$$

From the right hand side of Eq. (4) together with the modified version of Shannon's inequality for discrete variables [27]

$$\frac{1}{2}\ln\left[2\pi e\left(\Delta n^2 + \frac{1}{12}\right)\right] \geq S_N \geq S_N^{(2)}, \tag{5}$$

we find the alternative lower bound for the purity entropy

$$S^{(2)} \geq \frac{1}{e\pi}\exp(2S_N) - \frac{1}{6} \geq \frac{1}{e\pi}\exp\left(2S_N^{(2)}\right) - \frac{1}{6}. \tag{6}$$

For $S_N^{(2)} > \ln(e\pi/6)/2 \approx 0.18$ this bound is positive and for $S_N^{(2)} \gtrsim 1.25$ it is a stricter lower bound than the trivial relation (3).

### 2.2 Upper bound on the purity entropy

In order to derive an upper bound on the purity entropy we make use of the fact that the quantum state $\rho$ for a non-interacting fermionic system in any dimension is completely determined by its single-particle correlations and has a Gaussian form. This applies in particular to all eigenstates of free-fermion Hamiltonians and to all time-evolved states under such Hamiltonians if the initial state is Gaussian. Since we assume, furthermore, total particle number conservation, $\rho$ can be represented as [28–30]

$$\rho = \frac{1}{Z}\exp\left(-\sum_{mn} c_m^\dagger C_{mn} c_n\right), \tag{7}$$

where $c_m(c_m^\dagger)$ are the fermionic annihilation (creation) operators at lattice site $m$. Here $\mathbf{C}$ is a Hermitian matrix which is determined entirely by the matrix $\mathbf{f}$ of (normal) single-particle correlations

$$f_{mn} = \left\langle c_m^\dagger c_n \right\rangle = \text{tr}\left(\rho \, c_m^\dagger c_n\right) = \left[\frac{\mathbf{1}}{\mathbf{1} + e^{\mathbf{C}}}\right]_{mn}, \tag{8}$$

and $Z = \text{tr}\left\{\exp\left(-\sum_{nm} c_n^\dagger C_{nm} c_m\right)\right\}$.

We will now show that the particle number fluctuations $\Delta n^2$ in a partition are bounded from above by the Rényi number entropy $S_N^{(2)}$. Making use again of relation (4) this will then result in an upper bound on the purity entropy in terms of $S_N^{(2)}$. To do so, it is useful to introduce the moment generating function of the total particle number $\hat{N}$ in the partition [15]

$$\chi(\theta) \equiv \left\langle e^{i\theta\hat{N}} \right\rangle = \text{tr}\left\{\rho \, e^{i\theta\hat{N}}\right\} = \sum_n p(n) \, e^{in\theta}, \tag{9}$$

whose Fourier coefficients are the probabilities $p(n)$ to find $n$ particles in the subsystem. For Gaussian fermionic states, the generating function can be written as a determinant [31]

$$\chi(\theta) = \det\left[\mathbf{1} + (e^{i\theta} - 1)\frac{\mathbf{1}}{\mathbf{1} + e^{\mathbf{C}}}\right].$$

Making use of Parsevals theorem one then finds

$$
\begin{aligned}
\sum_n p^2(n) &= \frac{1}{2\pi}\int_0^{2\pi} d\theta \, |\chi(\theta)|^2 = \frac{1}{2\pi}\int_0^{2\pi} d\theta \left|\det\left[\mathbf{1} + (e^{i\theta} - 1)\frac{\mathbf{1}}{\mathbf{1} + e^{\mathbf{C}}}\right]\right|^2 \\
&= \frac{1}{2\pi}\int_0^{2\pi} d\theta \, \frac{\det\left(\mathbf{1} + 2e^{\mathbf{C}}\cos\theta + e^{2\mathbf{C}}\right)}{\det^2\left(\mathbf{1} + e^{\mathbf{C}}\right)} = \frac{1}{2\pi}\int_0^{2\pi} d\theta \, \det(\mathbf{1} - \mathbf{G} + \mathbf{G}\cos\theta).
\end{aligned}
\tag{10}
$$

In the last equation we introduced the matrix

$$\mathbf{G} = \frac{2e^{\mathbf{C}}}{(1 + e^{\mathbf{C}})^2} = 2\,\mathbf{f}(1 - \mathbf{f}) \leq \frac{1}{2}\mathbf{1}. \tag{11}$$

We see that the argument in the last line of Eq. (10) is a positive-definite matrix. Thus we can apply the arithmetic-geometric inequality to get an upper bound on $\det(\mathbf{1} - \mathbf{G} + \mathbf{G}\cos\theta)$. Denoting the lattice size as $M$ we find

$$\sum_n p^2(n) \leq \frac{1}{2\pi}\int_0^{2\pi} d\theta \left[1 + \frac{(\cos\theta - 1)\text{tr}(\mathbf{G})}{M}\right]^M \rightarrow \frac{1}{2\pi}\int_0^{2\pi} d\theta \, \exp\left[(\cos\theta - 1)\text{tr}(\mathbf{G})\right], \tag{12}$$

where the last step holds in the thermodynamic limit $M \rightarrow \infty$. The integral can be calculated elementary in terms of the modified Bessel function of the first kind $I_0(x)$ resulting in

$$\sum_n p^2(n) \leq \exp(-\text{tr}(\mathbf{G})) I_0(\text{tr}(\mathbf{G})). \tag{13}$$

Furthermore, we see from Eq. (11) that the trace of the matrix $\mathbf{G}$ gives the fluctuations of the total particle number

$$\text{tr}(\mathbf{G}) = 2\,\text{tr}\left(\mathbf{f}(1 - \mathbf{f})\right) = 2\,\Delta n^2. \tag{14}$$

Combined with Eq. (13) we therefore find for the number entropy

$$S_N^{(2)} = -\ln\sum_n p^2(n) \geq 2\Delta n^2 - \ln\left(I_0\left(2\Delta n^2\right)\right). \tag{15}$$

Using the asymptotic expansion of the modified Bessel function in the limit of large $\Delta n^2$, this expression can be simplified to

$$S_N^{(2)} \geq \frac{1}{2} \ln\left(4\pi\Delta n^2\right). \tag{16}$$

In the opposite limit of small $\Delta n^2$ the contribution of the modified Bessel function in Eq. (15) can be neglected and we find instead

$$S_N^{(2)} \geq 2\Delta n^2. \tag{17}$$

Now making use of the left hand side of the inequality in Eq. (4), we eventually arrive at an upper bound on the purity entropy in terms of the Rényi number entropy. This bound can be written explicitly in the two limiting cases of either small values of $\Delta n^2$

$$S^{(2)} \lesssim (2\ln 2) S_N^{(2)}, \tag{18}$$

or large values of $\Delta n^2$

$$S^{(2)} \lesssim \frac{\ln 2}{\pi} \exp\left(2S_N^{(2)}\right). \tag{19}$$

Eqs. (18,19) and (6) are the main results of our paper. They show that the entanglement quantified by the logarithm of the purity entropy $S^{(2)}$ is bounded both from below *and* above by the number entropy. As a consequence, a growth of entanglement in free fermionic systems is always accompanied by a logarithmically slower growth of the number entropy.

## 3   Time evolution of entropies in free fermionic systems

Next, we illustrate our results by considering one-dimensional tight-binding models of non-interacting fermions. By applying a Jordan-Wigner transformation, these models can alternatively also be seen as spin-1/2 XX chains. We discuss free fermions without disorder in Sec. 3.1, with potential disorder leading to Anderson localization in Sec. 3.2, and with bond (off-diagonal) disorder resulting in a critical system in Sec. 3.3.

The Hamiltonian for all these systems has the same structure

$$H = -\sum_{j=1}^{L-1} J_j(\hat{c}_j^\dagger \hat{c}_{j+1} + h.c.) + \sum_{j=1}^{L} D_j \hat{c}_j^\dagger \hat{c}_j, \tag{20}$$

with hopping amplitudes $J_j$, onsite potentials $D_j$, system size $L$, and open boundary conditions. In the following, we always consider the Rényi and number entropies for a partition of size $l = L/2$. We are interested in the time evolution of the entanglement entropies following a quantum quench starting from an initial state $|\Psi_{\mathrm{ini}}\rangle$ which is not an eigenstate of the Hamiltonian (20).

We use exact diagonalization (ED) methods to obtain the eigenvalues of the reduced density matrix $\rho(t)$, see Eq. (7), which is calculated from the time-evolved state $|\Psi(t)\rangle = \mathrm{e}^{-i\hat{H}t} |\Psi_{\mathrm{ini}}\rangle$ following Refs. [28–30]. From the eigenvalues $f_m$ of the correlation matrix $\mathbf{f}$ in Eq. (8), we can directly obtain all quantities of interest efficiently. This includes the von-Neumann and purity entropies

$$\begin{aligned} S &= -\sum_m f_m \ln(f_m) - \sum_m (1 - f_m) \ln(1 - f_m), \\ S^{(2)} &= -\sum_m \ln(1 - 2f_m(1 - f_m)), \end{aligned} \tag{21}$$

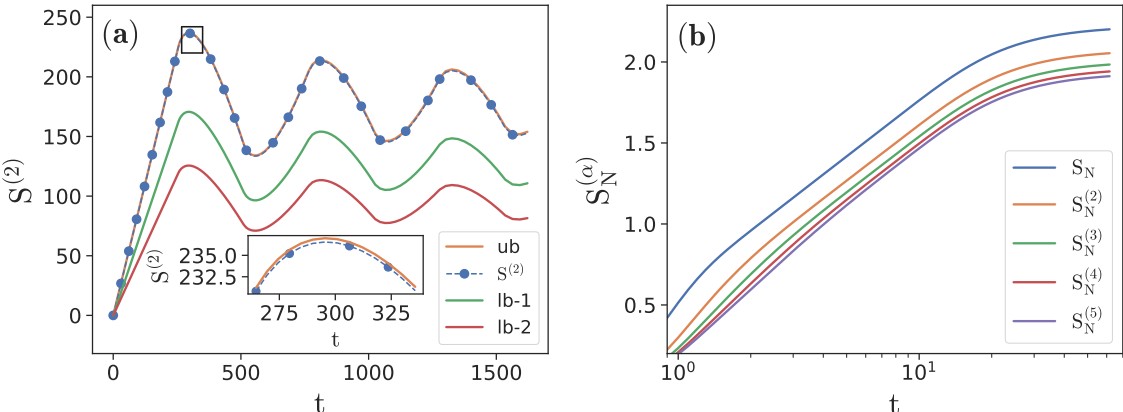

Figure 1: (a) $S^{(2)}(t)$ for a quench from the initial state (23) using the Hamiltonian (20) with $J_j = 1$, $D_j = 0$, and system size $L = 1024$. The purity entropy $S^{(2)}$ grows linearly until it reaches its maximum at $t \approx l/v$. The upper bound (ub), Eq. (19), and the two lower bounds from Eq. (6), (lb-1) $\exp(2S_N)/(e\pi) - 1/6$, and (lb-2) $\exp\left(2S_N^{(2)}\right)/(e\pi) - 1/6$, encapsule $S^{(2)}$. Note that the upper bound is tight, see inset. (b) Rényi number entropies $S_N^{(\alpha)}(t)$ for 20 Gaussian waves who are initially spaced at equal distances and whose width increases linearly in time, see Eq. (29), for $v = 1$. We find $S_N^{(\alpha)}(t) = \text{const} + \frac{1}{2}\ln t$. The saturation at long times is a finite-size effect.

as well as the number distribution which can be calculated from the corresponding characteristic function

$$p(n) = \frac{1}{l+1} \sum_{k=0}^{l} \exp\left(-i\frac{2\pi kn}{l+1}\right)\chi(k), \quad \chi(k) = \left\langle e^{i\frac{2\pi k}{l+1}\hat{N}} \right\rangle = \prod_m \left(1 + \left(e^{i\frac{2\pi k}{l+1}} - 1\right)f_m\right). \quad (22)$$

## 3.1 Free fermions without disorder

The case of fermions on a one-dimensional lattice has been studied extensively in the past, both analytically using conformal field theory [32] as well as numerically, see e.g. Ref. [23]. The conformal field theory results show that the von-Neumann entropy as well as all Rényi entropies increase linearly in time in the thermodynamic limit although with a slope which is non-universal. Here we choose a density-wave state with a fermion on every second site,

$$|\Psi_{\text{ini}}\rangle = \prod_{j=1}^{l} \hat{c}_{2j}^\dagger |0\rangle, \quad (23)$$

as initial state. Note, however, that the results are qualitatively the same for any generic initial product state. The numerical results in Fig. 1(a) show that $S^{(2)}(t)$ increases linearly in time until the particle-hole pairs created by the quench reach the boundaries of the partition of size $l = L/2$. This happens for times $t \sim l/v \approx l/2$ [23, 32] where $v \approx 2$ is the velocity of the excitations. For times $t > l/v$ boundary effects dominate the dynamics and $S^{(2)}$ is oscillating around an average value which depends on the size of the partition. Fig. 1(a) confirms that the lower and upper bounds obtained here are valid for all times, including long times where boundary effects dominate. The upper bound (19), in particular, is a very tight bound for all times in this case, see the inset of Fig. 1(a).

Based on the bounds and verified by the numerical results above we find that the Rényi number entropy grows as $S_N^{(2)}(t) \sim \ln t$ for free fermions on a one-dimensional lattice without

disorder. This logarithmic growth can be understood as follows: Consider a quench in a half-filled system where at long times each arrangement of particles has approximately the same probability. Then for a system of size $2L$ we have $L$ particles. If we cut the system in two halfs of size $L$, the probability to find $k$ particles in one half is given by

$$p(k, L) = \frac{\binom{L}{k}\binom{L}{L-k}}{\binom{2L}{L}}. \tag{24}$$

For $1 \ll k < L$ we can approximate this distribution by a normal distribution

$$\tilde{p}(k, L) = \frac{2}{\sqrt{L\pi}} \exp\left[-\frac{4}{L}\left(k - \frac{L}{2}\right)^2\right]. \tag{25}$$

We can now obtain the Rényi number entropies by integrating over the continuous distribution

$$S_N^{(\alpha)}(L) \approx \frac{1}{1-\alpha} \ln\left(\int_{-\infty}^{\infty} dk\, \tilde{p}^{\,\alpha}(k, L)\right) = \ln\left[\frac{\sqrt{\pi}}{2\alpha^{1/(2-2\alpha)}}\right] + \frac{1}{2}\ln L. \tag{26}$$

The von-Neumann number entropy can be obtained by

$$S_N(L) = \lim_{\alpha \to 1} S_N^{(\alpha)}(L) \approx \frac{1}{2}\left(1 + \ln\left[\frac{L\pi}{4}\right]\right). \tag{27}$$

If we now consider excitations which spread ballistically $\sim vt$ then we have regions of size $2L \sim vt$ in which each arrangement of particles has approximately equal probability. Putting this into the results for the number entropy and the Rényi number entropies we obtain the final result

$$S_N^{(\alpha)}(t) = \text{const} + \frac{1}{2}\ln t, \tag{28}$$

which includes the von-Neumann case ($\alpha \to 1$). Note that the constants do depend on the microscopic details of the model but are monotonically decreasing with $\alpha$, see Eq. (26). Thus we conclude that all Rényi number entropies behave the same qualitatively and $S_N^{(\alpha)} > S_N^{(\alpha+1)}$.

An alternative perspective to understand the logarithmic spreading—more closely related to the numerical simulations— can be obtained by considering Gaussian waves

$$|\Psi_i(x, t)|^2 = \frac{1}{\sqrt{4\pi vt^2}} \exp\left(-\frac{(x - x_i)^2}{4vt^2}\right), \tag{29}$$

with initial positions $x_i$ spread evenly along a line. Here $v$ is a constant and the width of the Gaussian wave is increasing linearly in time. The probability to find the particle $i$ at $x > 0$ is then given by

$$P(x_i, t) = \int_0^{\infty} |\Psi_i(x, t)|^2 = \frac{1}{2}\left(1 + \text{erf}\left(\frac{x_i}{2\sqrt{vt^2}}\right)\right). \tag{30}$$

If we have $N$ particles in total then the probability to find $k$ at $x > 0$ is

$$p(k, t) = \sum_{n_i \in \{0, 1\}}' \prod_{i=1}^{N} [P(x_i, t)]^{n_i} [1 - P(x_i, t)]^{1-n_i}. \tag{31}$$

The sum $\sum'$ is over all permutations of the $\{n_i\}$ and has to be evaluated with the constraint $\sum_{i=1}^{N} n_i = k$. It can be directly evaluated if $N$ is not too large. Results for $N = 20$ particles are shown in Fig. 1 (b) and confirm Eq. (28).

### 3.2 Anderson Localization

Static potential disorder in an isolated quantum system of non-interacting particles can induce Anderson localization (AL), defined as the absence of particle diffusion [33]. For one and two dimensions, Anderson localization occurs for any strength of disorder $D$ [34]. For a one-dimensional system we can extract the localization length $\xi$ in dependence of energy $\epsilon$ and disorder strength $D$ using a transfer matrix approach as described in [35]. If we quench a one-dimensional system with potential disorder we thus expect that for times $t \gg \xi/v$ both the number and configurational entropies will stop increasing.

To study the Anderson case numerically, we set the hopping in Eq. (20) uniformly to $J_j = 1$ and draw random values for the potential from a box distribution $D_j \in [-D/2, D/2]$. We now quench the system from initial random product states at half filling

$$|\Psi_{\text{ini}}\rangle = \prod_{j=1}^{L} \eta_j \hat{c}_j^\dagger |0\rangle, \tag{32}$$

where $\eta_j \in \{0, 1\}$ are random, and half-filling is imposed by requiring $\sum_j \eta_j = L/2$. The random product state will on average yield a state with energy $\epsilon = 0$ and we consider both a weakly disordered case, $D = 2$, and a strongly disordered case, $D = 20$. Since we consider systems large compared to the localization length we do not expect a qualitative difference in the time dependence of the purity entropies in the two cases: there will be an increase of entropy in time until a constant value is reached that does depend on the localization length $\xi$ but is independent of system size $L$ if $L \gg \xi$. Our numerical simulations, shown in Fig. 2(a-b), verify this behavior. Furthermore, they also verify the lower and upper bounds in terms of the number entropy. Plotted are the average Renyi entropy $\overline{S^{(2)}}$ as well as the lower and upper bounds proportional to the averaged exponential of the number entropy $\overline{\exp\{2S_N^{(2)}\}}$. It should be noted, however, that the bounds hold for every individual disorder realization. We note, furthermore, that $S^{(2)}$ is bounded quite tightly from below by $S_N^{(2)}$ for small values of $S^{(2)}$, see Fig. 2(b). Here the trivial lower bound, Eq. (3), is the better bound since both the purity entropy and the corresponding number entropy are quite small in the localized regime.

### 3.3 Bond disorder

As the third example, we consider the Hamiltonian (20) with bond or off-diagonal disorder, where the hopping amplitudes are drawn from a box disorder distribution, $P(J) = \text{const}$ with $J_j \in (0, 1]$. It is known that this system in the thermodynamic limit is at an infinite randomness fixed point [36–39]. The mean localization length scales as $\xi_{\text{loc}}(\epsilon) \sim |\ln(\epsilon)|^\Psi$ with $\Psi = 1/2$ being the critical exponent. The system therefore shows a localization-delocalization transition as a function of energy for $\epsilon \to 0$. The entanglement dynamics of this model has been investigated previously in Ref. [23] and of the related transverse Ising chain in Ref. [22].

Starting from the random half-filled state in Eq. (32) we show in Fig. 2(c) converged results for the time evolution of $S^{(2)}(t)$ obtained from exact diagonalizations of a system with $L = 1024$ lattice sites over very long times. One notices an extremely slow, but monotonic double-logarithmic increase of $S^{(2)}$ in time consistent with the results in Ref. [23]. The data provide, furthermore, verification of the upper and lower bounds for $S^{(2)}$ in terms of the number entropy derived in Sec. 2. We conclude that in the critical bond disordered case the number entropy in the thermodynamic limit also grows without bounds but extremely slowly, $S_N^{(2)} \sim \ln\ln\ln t$.

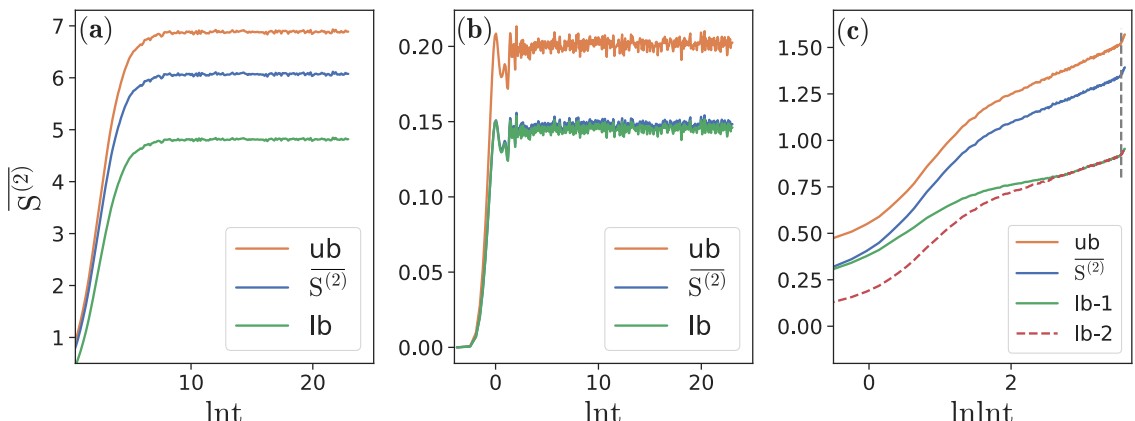

Figure 2: (a-b) $\overline{S^{(2)}}(t)$ for potential disorder after a quench starting from the half-filled random product state (32) averaged over 2000 disorder realizations and initial states. While all bounds hold for individual disorder realizations, we have plotted disorder averages, i.e. $\overline{S^{(2)}}$ and $\overline{\exp\left(2S_N^{(2)}\right)}$. (a) In the weakly disordered case $D = 2$ for $L = 512$ sites, the upper bound (ub), $(\ln 2/\pi)\overline{\exp\left(2S_N^{(2)}\right)}$ is quite tight for the regime in which $\overline{S^{(2)}}(t)$ grows. The lower bound (lb) shown is $\overline{\exp(2S_N)}/(e\pi) - 1/6$. (b) For strong disorder, $D = 20$ and $L = 128$ sites, the entanglement remains very small and Eq. (18), $2\ln(2)\overline{S_N^{(2)}}$, is the better upper bound (ub). For the same reason $\overline{S_N^{(2)}}$ (lb) is a better lower bound. (c) $\overline{S^{(2)}}(t)$ for bond disorder after a quench starting from the half-filled random product state (32) for $L = 1024$ sites, averaged over 20000 disorder realizations and initial states. (ub) corresponds to $(\ln 2/\pi)\overline{\exp\left(2S_N^{(2)}\right)}$, (lb-1) to the maximum of $\overline{S_N^{(2)}}$ and $\overline{\exp(2S_N)}/(\pi e) - 1/6$ and (lb-2) to $\overline{\exp(2S_N)}/(\pi e) - 1/6$. We confirm that $\overline{S^{(2)}} \sim \ln \ln t$. At long times, $\overline{S^{(2)}}$ and the upper bound (ub) (19) based on the number entropy only differ by a constant shift. The grey dotted line signals the point in time where double precision is no longer sufficient to obtain reliable results, see also Ref. [23].

## 4 Conclusions

In this paper we have considered the entanglement properties of Gaussian states of non-interacting fermions with particle number conservation. We have proven that for any such system - with and without disorder, on arbitrary lattice geometries, and in arbitrary dimensions - the second Rényi entanglement entropy $S^{(2)}$ can be bounded from *above and below* by the corresponding number entropy $S_N^{(2)}$. Our result implies an asymptotic scaling $S^{(2)} \propto \exp\left(S_N^{(2)}\right)$, i.e., a growth of the entanglement entropy always implies a growth, albeit logarithmically slower, of the number entropy and vice versa. While the precise upper and lower bounds have been derived for $S^{(2)}$, all Rényi entropies are expected to show the same asymptotic scaling with time or length. The connection between a growth in the entanglement entropy and a logarithmic slower growth in the corresponding number entropy is thus expected to hold for all Rényi entanglement entropies including the von-Neumann entanglement entropy.

Apart from being of fundamental importance for our understanding of entanglement in fermionic systems with particle number conservation, the bounds derived here are also useful for experiments on cold atomic gases. In such systems a measurement of the particle-number

distribution function $p(n,t)$ is possible [10] allowing to obtain any Rényi number entropy. Determining the entire configurational entropy and thus the full entanglement entropy experimentally, on the other hand, remains an open issue. Here our results provide an avenue to obtain the asymptotic scaling of the entanglement entropy from $p(n,t)$ alone.

The strict bounds for the entanglement in terms of the number entropy are not only physically instructive. We have found numerical evidence that similar relations between entanglement and number entropies also exist for interacting fermionic systems with particle number conservation, with important implications for phenomena resulting from the interplay between disorder and interactions such as many-body localization. These questions are studied in a recent preprint [21].

# Acknowledgements

J.S. acknowledges support by the Natural Sciences and Engineering Research Council (NSERC, Canada) and by the Deutsche Forschungsgemeinschaft (DFG) via Research Unit FOR 2316. We are grateful for the computing resources and support provided by Compute Canada and Westgrid. M.K., R.U., and M.F. acknowledge financial support from the Deutsche Forschungsgemeinschaft (DFG) via SFB TR 185, project number 277625399. M. K. would like to thank J. Léonard and M. Greiner for hospitality and fruitful discussions.

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
