# Peer review of "Bounds on the entanglement entropy by the number entropy in non-interacting fermionic systems"

_SciPost Physics, doi:SciPost Phys. 8, 083 (2020)_

## Round 1 · Referee Report · Anonymous (Referee 1) · 2020-3-18

Strengths

1- The main results are analytical and coincise. 2- The results are experimentally relevant. 3- The topic is timely and interesting.

Weaknesses

1- The lack of some definitions and the lack of discussion for some details hinder the clarity of the paper. 2- The section for systems with quenched disorder needs a deeper and clearer discussion. 3- Missing bibliography.

Report

Entanglement is a fundamental quantity to understand quantum matter. Despite the copious amount of theoretical investigations, its non-local non-classical nature renders entanglement difficult to measure in the present experimental setups. Within the specific setting of free fermion theories with particle number conservation, the authors present a powerful double inequality which bounds entanglement from above and from below. Remarkably these bounds involve uniquely a quantity that has been already accessed experimentally, the so called number entropy. Thus, their results provide a new estimate for entanglement in many body systems.

I think the paper is very interesting and relevant, and deserves to be published once the following issues are addressed:

1) My main concern is about the disorder system section. From the discussion, it is obscure which disorder averaged quantity the authors are considered in their numerical analysis. Is it $\exp 2 \overline{S^{(2)}}$ or $\overline{\exp 2 S^{(2)}}$, or others? This is important, as in the experimental paper of Lukin et al. (Ref. [5] of the present version), the authors I remember consider $\overline{p(n)}$. But clearly: \begin{equation}\overline{p(n)}^2\neq \overline{p(n)^2}. \end{equation} I would like to see an explicit and detailed analysis concerning these points.

A minor issue in the same context is that it is not clear from which distribution the $J_j$ are drawn. I'm not sure if the results depend or not on the specific distribution $P(J)$ and I would specify it. While at equilibrium, the universality of the random singlet phase (the infinite randomness fixed point of the model considered) guarantees independence of physical properties with respect to the functional form of the bond coupling distribution $P(J)$, the same it is not clear to me to hold in the non-equilibrium setting. I would like also a comment on this point.

2) My second main concern is that the bibliography is very poor. Despite many quantities in the paper have been widely studied and investigated, references to these earlier works are missing. Just to mention, I point out that number entropy (the central object in this paper) has been copiously analyzed in the context of symmetry resolved entanglement spectroscopy (see papers of Goldstein et al., Laflorencie et al., Murciano et. al, etc. ). Similarly the particle number generating function (related to full-counting statistics) has been as well considered in relation to entanglement (see papers of Song et al, Calabrese et al., other papers of Klich, etc. ). Overall I would suggest a substantial revision and update of the bibliography.

3) Finally since $p(n)$ is a central quantity of the work, I would add its definition and some comments on it to improve the clarity of the work. In the manuscript is not trasparent how the overall number conservation induced naturally one to consider the probability $p(n)$. I would like to see this point developed in the resubmitted version.

I end by mentioning a typo in eq.(10) (a \mathbf missing).

Requested changes

1- Clarify the quantities involved and the discussion in the disorder section. 2- Add the bond coupling distribution used and a comment on the universality/non-universality of the dynamics changing distribution. 3- Substantial revision and update of the bibliography. 4- Add definition and further discussion on $p(n)$ in the introduction.

  • validity: high
  • significance: high
  • originality: good
  • clarity: ok
  • formatting: excellent
  • grammar: excellent

Author:  Jesko Sirker  on 2020-04-28  [id 809]

(in reply to Report 1 on 2020-03-18)

We would like to thank the referee for the very positive general judgement of our work.

1) We thank the referee for pointing out this important issue. First of all, we would like to emphasize that the bounds for the entanglement entropy in terms of the number entropy derived in our manuscript hold for every individual disorder realization. The plots shown in Fig.2 for the exponential of the number entropy were obtained by averaging the exponential, i.e. $\overline{\exp(2S_N^{(2)})}$. We have now added corresponding explanations in Sec. 3.2 and in the caption of Fig. 2.

For the bond disorder case, the values of the hopping strength J were obtained from a box distribution, P(J)=const for $J\in (0,1]$. We now added a corresponding note in Sec. 3.3. In a previous work by one of us (J.S.), universality when changing the initial state has been studied numerically but not with respect to the form of the distribution P(J). We do, however, expect that the entropies are universal with respect to P(J) also in the non-equilibrium setting, but a verification would require further extensive numerical work, which is outside the scope of our paper. The point we would like to stress once more is that our bounds hold for every individual realization, i.e. they are independent of the distribution or initial state. So the question of universality with respect to P(J) is not directly relevant for our work.

2) We thank the referee for this comment, which has also been made by referee 2. We do agree that we should have paid a bit more attention here. Besides the references to seminal work by Klich et al. and Song et al. already included in the first version of the manuscript we have now added some references to other relevant work, e.g. the early contributions by Schuch et al. in PRL 92 (2004) and PRA 70 (2004), where particle fluctuations were considered as a quantum ressource, and the paper by Calabrese et al. in EPL 2012 where the cumulant expansion for Renyi entropies is discussed. Furthermore, we added references to theoretical proposals and experiments on the detection of the entanglement entropy. We hope that with the addition of Refs. [2-6,12-14,18,19] the bibliography is now more adequate.

3) We thank the referee for this well taken comment. We have now added a definition of p(n) right after its first appearance in Eq.(1). Furthermore we have added a comment in Sec. 2.2. that p(n) can also be seen as the Fourier coefficients of the moment generating function.

Finally, We thank the referee for pointing out the typo in Eq. (10), which we have corrected.

---

## Round 1 · Referee Report · Anonymous (Referee 2) · 2020-4-10

Strengths

A. The derived inequalities relate the particle number distribution, which is easier to measure experimentally, to the interesting but harder to measure entanglement entropy.
B. The derived inequalities are also theoretically useful, by providing bounds on the particle number distribution entropy when the entanglement entropy can be calculated and vice versa.

Weaknesses

A. The relation of the results to previous equality relations between the particle number distribution and the entanglement entropy for both free fermions and CFTs needs much elaboration.
B. Missing details on the disordered case, especially the system sizes and time windows as compared to the localization length.

Report

Entanglement has been realized to play a central role in the study of many-body system, as a way to both study the physics of the system and to characterize its potential for quantum information processing. However, the measurement of entanglement in many-body system is a significant challenge. The present work addresses this issue by giving analytic bounds on the second entanglement Renyi entropy in terms of the second Renyi entropy of the particle distribution. One of these bounds is general, the other specific to noninteracting fermionic systems. These bounds seem to be useful theoretically (since they involve the entanglement entropy and the exponent of the particle number entropy, they actually seem to be more useful in binding the latter in terms of the former rather than the other way around, a point the Authors should perhaps stress more). Moreover, the particle number distribution is readily measurable in cold atom systems, making the relation experimentally useful as well. For these reasons I think this is an important contribution, which is also nicely presented. However, before recommending its publication there are several points which need to be addressed.

Before getting to these points let me note that, after preparing a first draft of this report, I noticed another report has already been posted, whose conclusions largely overlap with those I have reached. Accordingly I have modified my report so as to only include points going beyond those previously made:

  1. As noted by the other Referee, related questions have been studied in previous works. I believe the issue here goes beyond expanding the bibliography. In particular, the string of works mentioned there (which should indeed all be cited), starting with Klich and Levitov [PRL 102, 100502 (2009)] derives an equality relation between entanglement entropy and particle number cumulants. Moreover, for large systems and long times, often the lowest (second) cumulant is sufficient. So a priori it seems that these older relations are potentially stronger than those derived here. A detailed discussion is therefore necessary.

  2. Similar issues apply to the more recent string of works on symmetry- or charge- resolved entanglement entropy, which generalizes the entanglement and particle number entropies studied in the current work. In these and subsequent studies some general results were given for CFTs of noninteracting and interacting fermions in both static and quench scenarios. The Authors should try to use these results to extract the quantities appearing in the current work and test the inequalities between them. The Authors may also consider generalizing the inequalities they derive to interacting 1D CFTs as well as to more general charge-resolved entropies.

  3. A full experimental test of the results would necessitate measuring independently the particle number distribution and the entanglement entropy. It would therefore be useful to mention other methods to experimentally extract the entanglement entropies, in particular using copies, as in the experiment Nature 528, 77 (2015), following theory works by the Zoller and additional groups; as well as using random time evolution, as in the experiment Phys. Rev. Lett. 124, 010504 (2020), following again the Zoller and other groups.

  4. Regarding disorder, the system sizes used for potential disorder do not seem to be given. Moreover, it would be useful to mention the critical exponent is \psi=1/2 for the bond-disordered system considered in this work. In addition, the resulting very slow divergence of the localization length makes it necessary to go to very large system sizes and times to observe the asymptotic behavior (here starting from the fixed point bond disorder distribution may help to some extent), especially since an extremely slow double-and triple-logarithmic behaviors are predicted. The Authors should comment on this.

Requested changes

1-2. The relation between the presented inequalities and previous equality results between entanglement and charge fluctuations needs much elaboration. 3. Adding references on entanglement entropy measurement. 4. More details in the disordered case are needed, especially regarding the appropriateness of the employed system sizes and time windows.

  • validity: high
  • significance: good
  • originality: good
  • clarity: good
  • formatting: excellent
  • grammar: excellent

Author:  Jesko Sirker  on 2020-04-28  [id 810]

(in reply to Report 2 on 2020-04-10)

We also thank referee 2 for the generally positive judgement of our work. The relation between the entropies can indeed be seen from different perspectives. Bounding the entanglement entropy from above and below by the same quantity, i.e. the number entropy, provides on the one hand a tool to measure entanglement. On the other hand, the lower bound on the number entropy by the logarithm of the entanglement entropy has important implications in the context of many-body localization, i.e. for interacting systems. For interacting fermions the relation is however only a conjecture, while for non-interacting fermions we provide a strict proof in our manuscript. The conjectured bounds in the interacting case and their implications for MBL are discussed in a different publication of us [21]. We have added a corresponding comment and reference [21] in the conclusions.

  1. As far as the bibliography is concerned, a similar comment has been made by referee 1 and we refer to our answer above.

The referee raises however another interesting point, namely the relation between entanglement and particle number cumulants. Here, several relations have been derived in the past by Klich and Levitov, Sun et al., Calabrese and coworkers and others. In this respect we would like to make the following comments:

(i) As pointed out by the referee, Klich and Levitov have derived an exact asymptotic expression of the number entropy in terms of particle number cumulants and often a few terms in this expansion provide a rather good approximation. We have calculated the approximation $\pi^2/4 C_2$, following the above mentioned work, for the examples shown in Fig. 1 and Fig. 2. While the agreement is indeed good (see the attached figures), it is also clear that the truncation of the asymptotic expansion at lowest order is neither a strict lower nor a strict upper bound. So a single quantity, such as the second cumulant, is not sufficient for our purposes. Furthermore, the expansion is an asymptotic one and its convergence is not guaranteed.

(ii) Our goal was to find strict lower and strict upper bounds for $S^{(2)}$ in terms of the same single quantity, characterizing number fluctuations. Only in this way can we establish a strict relation between the scaling properties of entanglement and number fluctuations.

(iii) For non-interacting fermions it is possible to bound the Renyi entropy from above and below by the variance of the subsystem particle number, see also Eq. (4), instead of the number entropy. (We note that Song et al. [19] have derived a strict lower bound in terms of number cumulants but an upper bound is missing.) Thus it appears at first glance that number fluctuations and number entropy are equally well suited for bounding the entanglement Renyi entropy. However, we have found numerical evidence in Ref. [21] that the relation involving the number entropy seems to carry over to the case of interacting fermions, while this appears less clear for the relation in terms of the number fluctuations. We have now modified the text in the abstract and introduction to emphasize that our aim was to find both upper and lower bounds for the entanglement in terms of a single quantity characterizing number fluctuations.

  1. We thank the referee for these interesting comments and suggestions for future directions. We agree that investigating the connection to other entropic quantities like the mentioned charge-resolved entropies is a very interesting direction of research. Also, the extension to interacting systems described by CFTs would be of interest. They are, however, outside of the scope of the present paper. Our work is concerned with the relation between the - easily detectable - number entropy and the entanglement entropy of a free fermion system and we derived bounds of the latter in terms of the first. Moreover, the systems studied by us are mostly not conformally invariant. As stated by referee 1: "Remarably these bounds involve uniquely a quantity that has been already accessed experimentally... Thus their results provide a new estimate for entanglement in many body systems". We thus did not follow this suggestion by the referee. We do note, however, that we have performed numerical studies for interacting fermion systems in [21] and found numerical evidence that our relation between number entropy and Renyi entropy carries over to this case with a proper renormalization of prefactors.

  2. We thank the referee for pointing out these papers and for bringing up the point of an experimental test of the relation between the entropies. Our paper reports a purely theoretical relation for which we provide a rigorous derivation. Thus an experimental verification for free fermion systems may perhaps be only of limited interest. However for interacting fermion systems a similar relation can only be conjectured, as we have done in Ref. [21]. Thus we agree with the referee that for such systems an experimental test would be very interesting and important. We have now added a corresponding discussion to the conclusions. Since the measurement of the entanglement entropy is an important issue in the context of our work, we also added the mentioned references in the introduction.

  3. We thank the referee for these comments. We were indeed a bit careless in providing all necessary details about the simulations. We now give the system sizes used and also mention the critical exponent $\psi =1/2$. We agree with the referee that due to the very slow divergence of the localization length rather large system sizes are needed. We note that a detailed analysis of the required system sizes and time scales has already been presented by one of us in Ref.[23]. We now also added a comment about the simulations of the bond-disordered case, shown in Fig.2c. Here a system size of L=1024 was used and the data are converged in length in the considered time interval.

Attachment:

Figures.pdf

---

## Round 2 · Referee Report · Anonymous (Referee 1) · 2020-5-3

Report

The authors replied to all the points raised in a satisfactory and exhaustive way, and I believe the paper deserves publication in Scipost Physics.

---

## Round 2 · Referee Report · Anonymous (Referee 2) · 2020-5-11

Strengths

As in my previous report

Weaknesses

Weaknesses pointed out in previous reports have been addressed

Report

In my opinion the Authors have addressed well all points previously made by both Referees. Considering my previous assessment of the work, I would now recommend its publication.

Requested changes

None

---

## Round 2 · Author Response

First of all, we would like to thank both referees for the very
careful and thorough evaluation of our work and the helpful
comments. In the following we will provide a detailed and point by
point reply. We have modified our manuscript accordingly and followed
most of the suggestions made. The few cases where we deviated from the
recommendations are explained as well.

---

## Round 2 · List of Changes

List of changes:

1) Abstract: "tightly bound---from above and below---by the"

2) Refs. [2-6] and discussion of experimental techniques to measure entanglement added.

3) Below Eq. (1): Definition of p(n) added.

4) Refs. [12-14] added.

5) Discussion of cumulants added at the bottom of page 2. Refs. [18,19] added.

7) Discussion of consequences for the interacting case added at the top of page 3. Ref. [21] added.

8) Below Eq. (9): p(n) defined as Fourier coefficients of the moment generating function.

9) Sec. 3.2 and caption of Fig. 2: Explanations about the disorder average added.

10) Sec 3.3.: Clarified that the amplitudes are drawn from a box distribution.

11) Sec 3.3.: Critical exponent Psi=1/2 specified.

12) Sec. 3.3.: System size added, clarified that results are converged in the time window shown.

13) Conclusions: Discussion about the relevance of the results for systems with interactions and disorder added.

---

## Editorial Decision

published